# DECOUPLED ADAPTATION FOR CROSS-DOMAIN OBJECT DETECTION

**Junguang Jiang, Baixu Chen, Jianmin Wang, Mingsheng Long**✉

School of Software, BNRist, Tsinghua University, China

{jjg20,chenbx18}@mails.tsinghua.edu.cn, {jimwang,mingsheng}@tsinghua.edu.cn

## ABSTRACT

Cross-domain object detection is more challenging than object classification since multiple objects exist in an image and the location of each object is unknown in the unlabeled target domain. As a result, when we adapt features of different objects to enhance the transferability of the detector, the features of the foreground and the background are easy to be confused, which may hurt the discriminability of the detector. Besides, previous methods focused on category adaptation but ignored another important part for object detection, *i.e.*, the adaptation on bounding box regression. To this end, we propose *D-adapt*, namely Decoupled Adaptation, to decouple the adversarial adaptation and the training of the detector. Besides, we introduce a bounding box adaptor to improve the localization performance. Experiments show that *D-adapt* achieves state-of-the-art results on four cross-domain object detection tasks and yields 17% and 21% relative improvement on benchmark datasets *Clipart1k* and *Comic2k* in particular.

## 1 INTRODUCTION

The object detection task has aroused great interest due to its wide applications. In the past few years, the development of deep neural networks has boosted the performance of object detectors [33; 15; 41]. While these detectors have achieved excellent performance on the benchmark datasets [11; 31], object detection in the real world still faces challenges from the large variance in viewpoints, object appearance, backgrounds, illumination, image quality, *etc*. Such domain shifts have been observed to cause significant performance drop [8]. Thus, some work uses *domain adaptation* [39] to transfer a detector from a source domain, where sufficient training data is available, to a target domain where only unlabeled data is available [8; 43]. This technique successfully improves the performance of the detector on the target domain. However, the improvement of domain adaptation in object detection remains relatively mild compared with that in object classification.

The inherent challenges come from three aspects. *Data challenge*: what to adapt in the object detection task is unknown. Instance feature adaptation in the object level (Figure 1(a)) might confuse the features of the foreground and the background since the generated proposals may not be true objects and many true objects might be missing (Figure 5). Global feature adaptation in the image level (Figure 1(b)) is likely to mix up features of different objects since each input image of detection has multiple objects. Local feature adaptation in the pixel level (Figure 1(c)) can alleviate domain shift when the shift is primarily low-level, yet it will struggle when the domains are different at the semantic level. *Architecture challenge*: while the above adaptation methods introduce domain discriminators and gradient reverse layers [12] into the detector architecture to encourage domain-invariant features, the discriminability of features might get deteriorated [6; 5], which will greatly influence the localization and the classification of the detectors. Besides, where to place these modules in the detection architecture has a great impact on the final performance but is a little tricky. Therefore, the scalability of these methods to different detection architectures is not so satisfactory. *Task challenge*: object detection is a multi-task learning problem, consisting of both classification and localization. Yet previous adaptation algorithms mainly explored the category adaptation, and it's still difficult to obtain an adaptation model suitable for different tasks at the same time.

To overcome these challenges, we propose a general framework – *D-adapt*, namely *Decoupled Adaptation*. Since adversarial alignment directly on the features of the detector might hurt its dis-

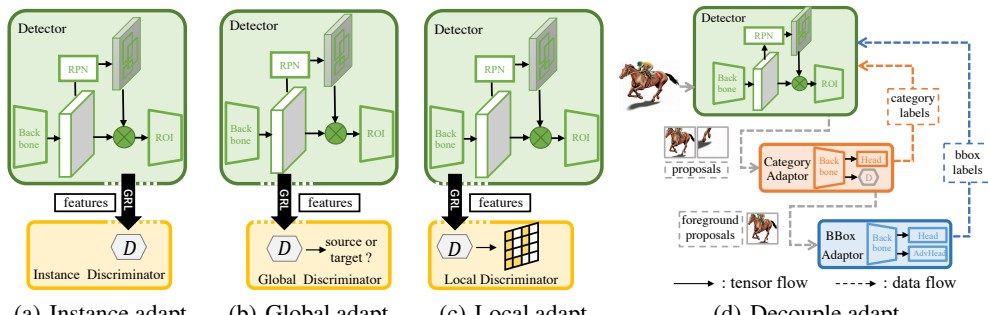

(a) Instance adapt    (b) Global adapt    (c) Local adapt    (d) Decouple adapt

Figure 1: Comparisons among techniques. Most previous methods can be categorized into instance adaptation [8], global adaptation [58], or local adaptation [43], which perform adaptation on the features of the detector. In decoupled adaptation, the adaptors are decoupled from the detector, and different adaptors are also decoupled. Decouple means that different parts have independent model parameters, independent input data distributions and independent training losses. Different parts are coordinated into some relationships through data rather than gradients, e.g., different adaptors form a cascading relationship while the detector and the adaptors form a self-feedback relationship.

criminability (architecture challenge), we decouple the adversarial adaptation from the training of the detector by introducing a parameter-independent category adaptor (see Figure 1(d)). To tackle the task challenge, we introduce another bounding box adaptor that's decoupled from both the detector and the category adaptor. To tackle the data challenge, we propose to adjust the object-level data distribution for specific adaptation tasks. For example, in the category adaptation step, we encourage the input proposals to have IoU[1] close to 0 or 1 to better satisfy the low-density separation assumption, while in the bounding box adaptation step, we encourage the input proposals to have IoU between 0.5 and 1 to ease the optimization of the bounding box localization task.

The contributions of this work are summarized as three-fold. (1) We introduce D-adapt framework for cross-domain object detection, which is general for both two-stage and single-stage detectors. (2) We propose an effective method to adapt the bounding box localization task, which is ignored by existing methods but is crucial for achieving superior final performance. (3) We conduct extensive experiments and validate that our method achieves state-of-the-art performance on four object detection tasks, and yields 17% and 21% relative improvement on Clipart1k and Comic2k.

## 2   RELATED WORK

**Generic domain adaptation for classification.** Domain adaptation is proposed to overcome the distribution shift across domains. In the classification setting, most of the domain adaptation methods are based on Moment Matching or Adversarial Adaptation. Moment Matching methods [50; 36] align distributions by minimizing the distribution discrepancy in the feature space. Taking the same spirit as Generative Adversarial Networks [16], Adversarial Adaptation [12; 37] introduces a domain discriminator to distinguish the source from the target, then the feature extractor is encouraged to fool the discriminator and learn domain invariant features. However, directly applying these methods to object detection yields an unsatisfactory effect. The difficulty is that the image of object detection usually contains multiple objects, thus the features of an image can have complex multimodal structures [20; 58; 5], making the image-level feature alignment problematic [58; 20].

**Generic domain adaptation for regression.** Most domain adaptation methods designed for classification do not work well on regression tasks since the regression space is continuous with no clear decision boundary [22]. Some specific regression algorithms are proposed, including importance weighting [54] or learning invariant representations [40; 38]. RSD [7] defines a geometrical distance for learning transferable representations and disparity discrepancy [57] proposes an upper bound for the distribution distance in the regression problems. Yet previous methods are mainly tested on simple tasks while this paper extends domain adaptation to the object localization tasks.

**Domain adaptation for object detection.** DA-Faster [8] performs feature alignment at both image-level and instance-level. SWDA [43] proposes that strong alignment of the local features is more effective than the strong alignment of the global features. Hsu *et al.* [20] carries out center-

---

[1]The Intersection-over-Union between the proposals and the ground-truth instance.

aware alignment by paying more attention to foreground pixels. HTCN [5] calibrates the transferability of feature representations hierarchically. Zheng *et al.* [59] proposes to extract foreground regions and adopts coarse-to-fine feature adaptation. ATF [19] introduces an asymmetric tri-way approach to account for the differences in labeling statistics between domains. CRDA [53] and MCAR [58] use multi-label classification as an auxiliary task to regularize the features. However, although the auxiliary task of outputting domain-invariant features to fool a domain discriminator in most aforementioned methods can improve the transferability, it also impairs the discriminability of the detector. In contrast, we decouple the adversarial adaptation and the training of the detector, thus the adaptors could specialize in transfer between domains, and the detector could focus on improving the discriminability while enjoying the transferability brought by the adaptors.

**Self-training with pseudo labels.** Pseudo-labeling [30], which leverages the model itself to obtain labels on unlabeled data, is widely used in self-training. To generate reliable pseudo labels, temporal ensembling [29] maintains an exponential moving average prediction for each sample, while the mean-teacher [49] averages model weights at different training iterations to get a teacher model. Deep mutual learning [56] trains a pool of student models with supervisions from each other. FixMatch [47] uses the model's predictions on weakly-augmented images to generate pseudo-labels for the strongly-augmented ones. Unbiased Teacher [35] introduces the teacher-student paradigm to Semi-Supervised Object Detection (SS-OD). When some image-level labels exist, the performance can be further improved by encoding correlations between coarse-grained and fine-grained classes [55], employing noise-tolerant training strategies [13], or learning a mapping from weakly-supervised to fully-supervised detectors [24] in SS-OD. Recent works [21; 25; 26] utilize self-training in cross-domain object detection and take the most confident predictions as pseudo labels. MTOR [3] uses the mean teacher framework and UMT [10] adopts distillation and CycleGAN [60] in self-training. However, self-training suffers from the problem of *confirmation bias* [1; 4]: the performance of the student will be limited by that of the teacher. Although pseudo labels are also used in our proposed D-adapt, they are generated from adaptors that have independent parameters and different tasks from the detector, thereby alleviating the confirmation bias of the overly tight relationship in self-training.

## 3 PROPOSED METHOD

In supervised object detection, we have a labeled source domain $\mathcal{D}_s = \{(\mathbf{X}_s^i, \mathbf{B}_s^i, \mathbf{Y}_s^i)\}_{i=1}^{n_s}$, where $\mathbf{X}_s^i$ is the image, $\mathbf{B}_s^i$ is the bounding box coordinates, and $\mathbf{Y}_s^i$ is the categories. The detector $G^{\text{det}}$ is trained with $\mathcal{L}_s^{\text{det}}$, which consists of four losses in Faster RCNN [42]: the RPN classification loss $\mathcal{L}_{\text{cls}}^{\text{rpn}}$, the RPN regression loss $\mathcal{L}_{\text{reg}}^{\text{rpn}}$, the RoI classification loss $\mathcal{L}_{\text{cls}}^{\text{roi}}$ and the RoI regression loss $\mathcal{L}_{\text{reg}}^{\text{roi}}$,

$$\mathcal{L}_s^{\text{det}} = \mathbb{E}_{(\mathbf{X}_s, \mathbf{B}_s, \mathbf{Y}_s) \in \mathcal{D}_s} \mathcal{L}_{\text{cls}}^{\text{rpn}} + \mathcal{L}_{\text{reg}}^{\text{rpn}} + \mathcal{L}_{\text{cls}}^{\text{roi}} + \mathcal{L}_{\text{reg}}^{\text{roi}}. \tag{1}$$

In cross-domain object detection, there exists another unlabeled target domain $\mathcal{D}_t = \{\mathbf{X}_t^i\}_{i=1}^{n_t}$ that follows different distributions from $\mathcal{D}_s$. The objective of $G^{\text{det}}$ is to improve the performance on $\mathcal{D}_t$.

### 3.1 D-ADAPT FRAMEWORK

To deal with the architecture challenge mentioned in Section 1, we propose the D-adapt framework, which has three steps: (1) decouple the original cross-domain detection problem into several sub-problems (2) design adaptors to solve each sub-problem (3) coordinate the relationships between different adaptors and the detector.

Since adaptation might hurt the discriminability of the detector, we *decouple* the category adaptation from the training of the detector by introducing a parameter-independent category adaptor (see Figure 1(d)). The adaptation is only performed on the features of the category adaptor, thus will not hurt the detector's ability to locate objects. To fill the blank of regression domain adaptation in object detection, we need to perform adaptation on the bounding box regression. Yet feature visualization in Figure 6(c) reveals that features that contain both category and location information do not have an obvious cluster structure, and alignment might hurt its discriminability. Besides, the common category adaptation methods are also not effective on regression tasks [22], thus we *decouple* category adaptation and the bounding box adaptation to avoid their interfering with each other. Section 3.2 and 3.3 will introduce the design of category adaptor and box adaptor in details. In this section, we will assume that such two adaptors are already obtained.

To coordinate the adaptation on different tasks, we maintain a *cascading* relationship between the adaptors. In the cascading structure, the later adaptors can utilize the information obtained by the previous adaptors for better adaptation, e.g. in the box adaptation step, the category adaptor will select foreground proposals to facilitate the training of the box adaptor. Compared with the multi-task learning relationship where we need to balance the weights of different adaptation losses carefully, the cascade relationship greatly reduces the difficulty of hyper-parameter selection since each adaptor has only one adaptation loss. Since the adaptors are specifically designed for cross-domain tasks, their predictions on the target domain can serve as pseudo labels for the detector. On the other hand, the detector generates proposals to train the adaptors and higher-quality proposals can improve the adaptation performance (see Table 5 for details). And this enables the *self-feedback* relationship between the detector and the adaptors.

For a good initialization of this self-feedback loop, we first pre-train the detector $G^{\text{det}}$ on the source domain with $\mathcal{L}_s^{\text{det}}$. Using the pre-trained $G^{\text{det}}$, we can derive two new data distributions, the source proposal distribution $\mathcal{D}_s^{\text{prop}}$ and the target proposal distribution $\mathcal{D}_t^{\text{prop}}$. Each proposal consists of a crop of the image $\mathbf{x}^2$, its corresponding bounding box $\mathbf{b}^{\text{det}}$, predicted category $\mathbf{y}^{\text{det}}$ and the class confidence $\mathbf{c}^{\text{det}}$. We can annotate each source-domain proposal $\mathbf{x}_s \in \mathcal{D}_s^{\text{prop}}$ with a ground truth bounding box $\mathbf{b}_s^{\text{gt}}$ and category label $\mathbf{y}_s^{\text{gt}}$, similar to labeling each RoI in Fast RCNN [14], and then use these labels to train the adaptors. In turn, for each target proposal $\mathbf{x}_t \in \mathcal{D}_t^{\text{prop}}$, adaptors will provide category pseudo label $\mathbf{y}_t^{\text{cls}}$ and box pseudo label $\mathbf{b}_t^{\text{reg}}$ to train the RoI heads,

$$\mathcal{L}_t^{\text{det}} = \mathbb{E}_{(\mathbf{X}_t, \mathbf{b}_t^{\text{det}}, \mathbf{y}_t^{\text{cls}}, \mathbf{b}_t^{\text{reg}}) \in \mathcal{D}_t^{\text{prop}}} \mathcal{L}_{\text{cls}}^{\text{roi}}(\mathbf{X}_t, \mathbf{b}_t^{\text{det}}, \mathbf{y}_t^{\text{cls}}) + \mathcal{L}_{\text{cls}}^{\text{roi}}(\mathbf{X}_t, \mathbf{b}_t^{\text{reg}}, \mathbf{y}_t^{\text{cls}})$$
$$+ \mathbb{I}_{\text{fg}}(\mathbf{y}_t^{\text{cls}}) \cdot \mathcal{L}_{\text{reg}}^{\text{roi}}(\mathbf{X}_t, \mathbf{b}_t^{\text{det}}, \mathbf{b}_t^{\text{reg}}), \tag{2}$$

where $\mathbb{I}_{\text{fg}}$ is a function that indicates whether it is a foreground class. Note that regression loss is activated only for foreground anchors. After obtaining a better detector by optimizing Equation 2, we can generate higher-quality proposals, which facilitate better category adaptation and bounding box adaptation. This process can iterate multiple times and the detailed optimization procedures are summarized in Algorithm 1.

Note that our D-adapt framework does not introduce any computational overhead in the inference phase, since the adaptors are independent of the detector and can be removed during detection. Also, D-adapt does not depend on a specific detector, thus the detector can be replaced by SSD [34], RetinaNet [32], or other detectors.

---

**Algorithm 1:** D-adapt Training Pipeline.

**input** : Source domain $\mathcal{D}_s$ and target domain $\mathcal{D}_t$, number of iterations $T$
**output:** Cross-domain object detector $G^{\text{det}}$

*initialize* the object detector $G^{\text{det}}$ by optimizing with $\mathcal{L}_s^{\text{det}}$;
**for** $t \leftarrow 1$ **to** $T$ **do**
    generate proposals $\mathcal{D}_s^{\text{prop}}$ and $\mathcal{D}_t^{\text{prop}}$ for each sample in $\mathcal{D}_s$ and $\mathcal{D}_t$ by $G^{\text{det}}$;
    **for** *each mini-batch in $\mathcal{D}_s^{prop}$ and $\mathcal{D}_t^{prop}$* **do**
        | train the category adaptor $G^{\text{cls}}$;
    **end**
    generate category label for each proposal in $\mathcal{D}_t^{\text{prop}}$;
    generate foreground proposals $\mathcal{D}_s^{\text{fg}}$ and $\mathcal{D}_t^{\text{fg}}$ from $\mathcal{D}_s^{\text{prop}}$ and $\mathcal{D}_t^{\text{prop}}$;
    **for** *each mini-batch in $\mathcal{D}_s^{fg}$ and $\mathcal{D}_t^{fg}$* **do**
        | train the bounding box adaptor $G^{\text{reg}}$;
    **end**
    generate bounding box label for each proposal in $\mathcal{D}_t^{\text{fg}}$;
    train the object detector $G^{\text{det}}$ by optimizing with $\mathcal{L}_t^{\text{det}}$;
**end**

---

### 3.2 CATEGORY ADAPTATION

The goal of category adaptation is to use labeled source-domain proposals $(\mathbf{x}_s, \mathbf{y}_s^{\text{gt}}) \in \mathcal{D}_s^{\text{prop}}$ to obtain a relatively accurate classification $\mathbf{y}_t^{\text{cls}}$ of the unlabeled target-domain proposals $\mathbf{x}_t \in \mathcal{D}_t^{\text{prop}}$. Some generic adaptation methods, such as DANN [12], can be adopted. DANN introduces a domain discriminator to distinguish the source from the target, then the feature extractor tries to learn domain-invariant representations to fool the discriminator, which will enlarge the decision boundaries between classes on the unlabeled target domain. However, the above adversarial alignment might fail due to the data challenge – *the input data distribution doesn't satisfy the low-density separation assumption well*, i.e., the Intersection-over-Union of a proposal and a foreground instance may be any value between 0 and 1 (see Figure 2(a)) and explicit task-specific boundaries between classes hardly exist, which will impede the adversarial alignment [22]. Recall that in standard object detection, proposals with IoU between $0.3$ and $0.7$ will be removed to discretize the input space and ease the optimization of the classification. Yet it can hardly be used in the domain adaptation problem since we cannot obtain ground truth IoU for target proposals.

---

[2]We use uppercase letters to represent the whole image, lowercase letters to represent an instance of object.

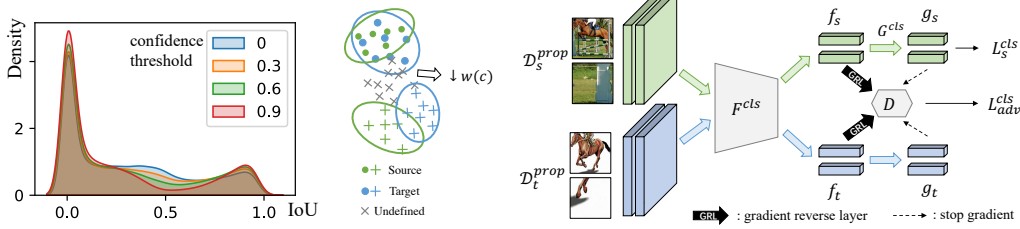

(a) IoU distribution of proposals   (b) Discretization   (c) Architecture of the category adaptor

Figure 2: Category adaptation (best viewed in color). **(a)** The IoU distribution of the proposals from Foggy Cityscapes. When we increase the confidence threshold from 0 to 0.9, undefined proposals (proposals with IoU between 0.3 and 0.7) will decrease. **(b)** Proposals with lower confidence will be assigned a lower weight in the adaptation. **(c)** The discriminator $D$ is trained to separate the source-domain proposals from the target-domain proposals for each class independently, while the feature extractor $F^{\mathrm{cls}}$ is encouraged to fool $D$.

To overcome the data challenge, we use the confidence of each proposal to discretize the input space, i.e., when a proposal has a high confidence $\mathbf{c}^{\mathrm{det}}$ being the foreground or background, it should have a higher weight $w(\mathbf{c}^{\mathrm{det}})$ in the adaptation, and vice versa (see Figure 2(b)). This will reduce the participation of proposals that are neither foreground nor background and improve the discreteness of the input space in the sense of probability. Then the objective of the discriminator $D$ is,

$$\max_{D} \mathcal{L}^{\mathrm{cls}}_{\mathrm{adv}} = \mathbb{E}_{\mathbf{x}_s \sim \mathcal{D}^{\mathrm{prop}}_s} w(\mathbf{c}_s) \log[D(\mathbf{f}_s, \mathbf{g}_s)] + \mathbb{E}_{\mathbf{x}_t \sim \mathcal{D}^{\mathrm{prop}}_t} w(\mathbf{c}_t) \log[1 - D(\mathbf{f}_t, \mathbf{g}_t)], \quad (3)$$

where both the feature representation $\mathbf{f} = F^{\mathrm{cls}}(\mathbf{x})$ and the category prediction $\mathbf{g} = G^{\mathrm{cls}}(\mathbf{f})$ are fed into the domain discriminator $D$ (see Figure 2(c)). This will encourage features aligned in a conditional way [37], and thus avoid that most target proposals aligned to the dominant category on the source domain. The objective of the feature extractor $F^{\mathrm{cls}}$ is to separate different categories on the source domain and learn domain-invariant features to fool the discriminator,

$$\min_{F^{\mathrm{cls}}, G^{\mathrm{cls}}} \mathbb{E}_{(\mathbf{x}_s, \mathbf{y}^{\mathrm{gt}}_s) \sim \mathcal{D}^{\mathrm{prop}}_s} \mathcal{L}_{\mathbf{CE}}(G^{\mathrm{cls}}(\mathbf{f}_s), \mathbf{y}^{\mathrm{gt}}_s) + \lambda \mathcal{L}^{\mathrm{cls}}_{\mathrm{adv}}, \quad (4)$$

where $\mathcal{L}_{\mathbf{CE}}$ is the cross-entropy loss, $\lambda$ is the trade-off between source risk and domain adversarial loss. After obtaining the adapted classifier, we can generate category pseudo label $\mathbf{y}^{\mathrm{cls}}_t = G^{\mathrm{cls}} \circ F^{\mathrm{cls}}(\mathbf{x}_t)$ for each proposal $\mathbf{x}_t \in \mathcal{D}^{\mathrm{prop}}_t$.

## 3.3 BOUNDING BOX ADAPTATION

The objective of box adaptation is to utilize labeled source-domain foreground proposals $(\mathbf{x}_s, \mathbf{b}^{\mathrm{gt}}_s) \in \mathcal{D}^{\mathrm{fg}}_s$ to obtain bounding box labels $\mathbf{b}^{\mathrm{reg}}_t$ of the unlabeled target-domain proposals $\mathbf{x}_t \in \mathcal{D}^{\mathrm{fg}}_t$. Recall that in object detection, regression loss is activated only for foreground anchor and is disabled otherwise [14], thus we only adapt the foreground proposals when training the bounding box regressor. Since the ground truth labels of target-domain proposals are unknown, we use the prediction obtained in the category adaptation step, i.e. $\mathcal{D}^{\mathrm{fg}}_t = \{(\mathbf{x}_t, \mathbf{y}^{\mathrm{cls}}_t) \in \mathcal{D}^{\mathrm{prop}}_t | \mathbb{I}_{\mathrm{fg}}(\mathbf{y}^{\mathrm{cls}}_t)\}$.

Following RCNN [15], we adopt a class-specific bounding-box regressor, which predicts the bounding box regression offsets, $t^k = (t^k_x, t^k_y, t^k_w, t^k_h)$ for each of the $K$ foreground classes, indexed by $k$. On the source domain, we have the ground truth category and bounding box label for each proposal, thus we use the smooth $L_1$ loss to train the regressor,

$$\min_{F^{\mathrm{reg}}, G^{\mathrm{reg}}} \mathcal{L}^{\mathrm{reg}}_s = \mathbb{E}_{(\mathbf{x}_s, \mathbf{y}^{\mathrm{gt}}_s, \mathbf{b}^{\mathrm{gt}}_s, \mathbf{b}^{\mathrm{det}}_s) \sim \mathcal{D}^{\mathrm{fg}}_s} \sum_{i \in \{x, y, w, h\}} \mathrm{smooth}_{L_1}(t^u_i - v_i), \quad (5)$$

where $t = G^{\mathrm{reg}} \circ F^{\mathrm{reg}}(\mathbf{x}_s)$ is the regression prediction, $u = \mathbf{y}^{\mathrm{gt}}_s$ is ground truth category, $v$ is the ground truth bounding box offsets calculated from $\mathbf{b}^{\mathrm{gt}}_s$ and $\mathbf{b}^{\mathrm{det}}_s$. However, it's hard to obtain a satisfactory regressor with $\mathcal{L}^{\mathrm{reg}}_s$ on the target domain due to the domain shift.

Inspired by the lastest theory [57], we propose an IoU disparity discrepancy method. As shown in Figure 3(a), we train a feature generator network $F^{\mathrm{reg}}$ which takes proposal inputs, and two regressor networks $G^{\mathrm{reg}}$ and $G^{\mathrm{reg}}_{\mathrm{adv}}$ which take features from $F^{\mathrm{reg}}$. The objective of the adversarial

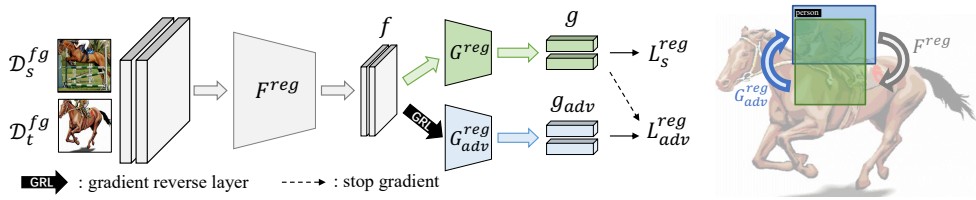

(a) Architecture of the bounding box adaptor      (b) Minimax on IoU

Figure 3: Bounding box adaptation (best viewed in color). Box adaptor has three parts: feature generator $F^{\text{reg}}$, regressor $G^{\text{reg}}$ and adversarial regressor $G_{\text{adv}}^{\text{reg}}$. $G_{\text{adv}}^{\text{reg}}$ learns to maximize the target disparity by moving two predicted boxes far from each other while $F^{\text{reg}}$ learns to minimize the target disparity by making two predicted boxes overlap as much as possible.

regressor network $G_{\text{adv}}^{\text{reg}}$ is to maximize its disparity with the main regressor on the target domain while minimizing the disparity on the source domain to measure the discrepancy across domains. Then the objective of the adversarial regressor is

$$\max_{G_{\text{adv}}^{\text{reg}}} \mathcal{L}_{\text{adv}}^{\text{reg}} = \mathbb{E}_{(\mathbf{x}_t, \mathbf{y}_t^{\text{cls}}) \sim \mathcal{D}_t^{\text{fg}}} \text{smooth}_{L_1}(G_{\text{adv}}^{\text{reg}} \circ F^{\text{reg}}(\mathbf{x}_t)^{\mathbf{y}_t^{\text{cls}}}, G^{\text{reg}} \circ F^{\text{reg}}(\mathbf{x}_t)^{\mathbf{y}_t^{\text{cls}}})$$

$$- \mathbb{E}_{(\mathbf{x}_s, \mathbf{y}_s^{\text{gt}}) \sim \mathcal{D}_s^{\text{fg}}} \text{smooth}_{L_1}(G_{\text{adv}}^{\text{reg}} \circ F^{\text{reg}}(\mathbf{x}_s)^{\mathbf{y}_s^{\text{gt}}}, G^{\text{reg}} \circ F^{\text{reg}}(\mathbf{x}_s)^{\mathbf{y}_s^{\text{gt}}}). \tag{6}$$

Note that $\text{smooth}_{L_1}$ on the source domain is only defined on the box corresponding to the ground truth category $\mathbf{y}_s^{\text{gt}}$ and that on the target domain is only defined on the box associated with the predicted category $\mathbf{y}_t^{\text{cls}}$. Equation 6 guides the adversarial regressor to predict correctly on the source domain while making as many mistakes as possible on the target domain (Figure 3(b)). Then the feature extractor $F^{\text{reg}}$ is encouraged to output domain-invariant features to decrease domain discrepancy,

$$\min_{F^{\text{reg}}} \mathcal{L}_s^{\text{reg}} + \eta \mathcal{L}_{\text{adv}}^{\text{reg}}, \tag{7}$$

where $\eta$ is the trade-off between source risk and adversarial loss. After obtaining the adapted regressor, we can generate box pseudo label $\mathbf{b}_t^{\text{reg}} = G^{\text{reg}} \circ F^{\text{reg}}(\mathbf{x}_t)$ for each proposal $\mathbf{x}_t \in \mathcal{D}_t^{\text{fg}}$.

## 4 EXPERIMENTS

### 4.1 DATASETS

Following six object detection datasets are used: Pascal VOC [11], Clipart [21], Comic [21], Sim10k [23], Cityscapes [9] and FoggyCityscapes [44]. Pascal VOC contains 20 categories of common real-world objects and $16,551$ images. Clipart contains 1k images and shares 20 categories with Pascal VOC. Comic2k contains 1k training images and 1k test images, sharing 6 categories with Pascal VOC. Sim10k has $10,000$ images with $58,701$ bounding boxes of car categories, rendered by the gaming engine Grand Theft Auto. Both Cityscapes and FoggyCityscapes have 2975 training images and 500 validation images with 8 object categories. Following [43], we evaluate the domain adaptation performance of different methods on the following four domain adaptation tasks, VOC-to-Clipart, VOC-to-Comic2k, Sim10k-to-Cityscapes, Cityscapes-to-FoggyCityscapes, and report the mean average precision (mAP) with a threshold of 0.5.

### 4.2 IMPLEMENTATION DETAILS

**Stage 1: Source-domain pre-training.** In the basic experiments, Faster-RCNN [42] with ResNet-101 [17] or VGG-16 [46] as backbone is adopted and pre-trained the on the source domain with a learning rate of 0.005 for 12k iterations.

**Stage 2: Category adaptation.** The category adaptor has the same backbone as the detector but a simple classification head. It's trained for $10k$ iterations using SGD optimizer with an initial learning rate of 0.01, momentum 0.9, and a batch size of 32 for each domain. The discriminator $D$ is a three-layer fully connected networks following DANN [12]. $\lambda$ is kept 1 for all experiments. $w(\mathbf{c})$ is 1 when $\mathbf{c} > 0.5$ and 0 otherwise.

Table 1: Results from PASCAL VOC to Clipart (ResNet101).

| | aero | bcycle | bird | boat | bottle | bus | car | cat | chair | cow | table | dog | hrs | bike | prsn | plnt | sheep | sofa | train | tv | mAP |
|---|---|---|---|---|---|---|---|---|---|---|---|---|---|---|---|---|---|---|---|---|---|
| Source Only | 35.6 | 52.5 | 24.3 | 23.0 | 20.0 | 43.9 | 32.8 | 10.7 | 30.6 | 11.7 | 13.8 | 6.0 | 36.8 | 45.9 | 48.7 | 41.9 | 16.5 | 7.3 | 22.9 | 32.0 | 27.8 |
| DA-Faster [8] | 15.0 | 34.6 | 12.4 | 11.9 | 19.8 | 21.1 | 23.2 | 3.1 | 22.1 | 26.3 | 10.6 | 10.0 | 19.6 | 39.4 | 34.6 | 29.3 | 1.0 | 17.1 | 19.7 | 24.8 | 19.8 |
| BDC-Faster [43] | 20.2 | 46.4 | 20.4 | 19.3 | 18.7 | 41.3 | 26.5 | 6.4 | 33.2 | 11.7 | 26.0 | 1.7 | 36.6 | 41.5 | 37.7 | 44.5 | 10.6 | 20.4 | 33.3 | 15.5 | 25.6 |
| WST-BSR [27] | 28.0 | 64.5 | 23.9 | 19.0 | 21.9 | 64.3 | 43.5 | 16.4 | 42.0 | 25.9 | 30.5 | 7.9 | 25.5 | 67.6 | 54.5 | 36.4 | 10.3 | 31.2 | 57.4 | 43.5 | 35.7 |
| SWDA [43] | 26.2 | 48.5 | 32.6 | 33.7 | 38.5 | 54.3 | 37.1 | 18.6 | 34.8 | 58.3 | 17.0 | 12.5 | 33.8 | 65.5 | 61.6 | 52.0 | 9.3 | 24.9 | 54.1 | 49.1 | 38.1 |
| MAF [18] | 38.1 | 61.1 | 25.8 | **43.9** | 40.3 | 41.6 | 40.3 | 9.2 | 37.1 | 48.4 | 24.2 | 13.4 | 36.4 | 52.7 | 57.0 | **52.5** | 18.2 | 24.3 | 32.9 | 39.3 | 36.8 |
| SCL [45] | 44.7 | 50.0 | 33.6 | 27.4 | 42.2 | 55.6 | 38.3 | 19.2 | 37.9 | **69.0** | 30.1 | **26.3** | 34.4 | 67.3 | 61.0 | 47.9 | 21.4 | 26.3 | 50.1 | 47.3 | 41.5 |
| CRDA [53] | 28.7 | 55.3 | 31.8 | 26.0 | 40.1 | 63.6 | 36.6 | 9.4 | 38.7 | 49.3 | 17.6 | 14.1 | 33.3 | 74.3 | 61.3 | 46.3 | 22.3 | 24.3 | 49.1 | 44.3 | 38.3 |
| HTCN [5] | 33.6 | 58.9 | 34.0 | 23.4 | **45.6** | 57.0 | 39.8 | 12.0 | 39.7 | 51.3 | 21.1 | 20.1 | 39.1 | 72.8 | 63.0 | 43.1 | 19.3 | 30.1 | 50.2 | 51.8 | 40.3 |
| ATF [19] | 41.9 | **67.0** | 27.4 | 36.4 | 41.0 | 48.5 | 42.0 | 13.1 | 39.2 | 75.1 | 33.4 | 7.9 | 41.2 | 56.2 | 61.4 | 50.6 | **42.0** | 25.0 | 53.1 | 39.1 | 42.1 |
| Unbiased [35] | 30.9 | 51.8 | 27.2 | 28.0 | 31.4 | 59.0 | 34.2 | 10.0 | 35.1 | 19.6 | 15.8 | 9.3 | 41.6 | 54.4 | 52.6 | 40.3 | 22.7 | 28.8 | 37.8 | 41.4 | 33.6 |
| D-adapt | **56.4** | 63.2 | **42.3** | 40.9 | 45.3 | **77.0** | **48.7** | **25.4** | **44.3** | 58.4 | **31.4** | 24.5 | **47.1** | 75.3 | 69.3 | 43.5 | 27.9 | **34.1** | 60.7 | **64.0** | **49.0** |

| VOC→Clipart | VOC→Comic | Sim10k→Cityscapes | Cityscapes→Foggy Cityscapes |

Figure 4: Qualitative results on the target domain.

**Stage 3: Bounding box adaptation.** The box adaptor has the same backbone as the detector but a simple regression head (two-layer convolutions networks). The training hyper-parameters (learning rate, batch size, etc.) are the same as that of the category adaptor. $\eta$ is kept $0.1$ for all experiments. The input of the bounding box adaptor (the crops of objects) will be twice larger than the original predicted box, so that the bounding box adapter could access more location information.

**Stage 4: Target-domain pseudo-label training.** The detector is trained on the target domain for $4k$ iterations, with an initial learning rate of $2.5 \times 10^{-4}$ and reducing to $2.5 \times 10^{-5}$ exponentially.

The adaptors and the detector are trained in an alternative way for $T = 3$ iterations. We perform all experiments on public datasets using a 1080Ti GPU. **Code is available at https://github.com/thuml/Decoupled-Adaptation-for-Cross-Domain-Object-Detection**.

## 4.3 COMPARISON WITH STATE-OF-THE-ARTS

**Adaptation between dissimilar domains.** We first show experiments on dissimilar domains using the Pascal VOC Dataset as the source domain and Clipart as the target domain. Table 1 shows that our proposed method outperforms the state-of-the-art method by 6.9 points on mAP. Figure 4 presents some qualitative results in the target domain. We also compare with Unbiased Teacher [35], the state-of-the-art method in semi-supervised object detection, which generates pseudo labels on the target domain from the teacher model. Due to the large domain shift, the prediction from the teacher detection model is unreliable, thus it doesn't do well. In contrast, our method alleviates the confirmation bias problem by generating pseudo labels from different models (adaptors).

We also use Comic2k as the target domain, which has a very different style from Pascal VOC and a lot of small objects. As shown in Table 2, both image-level and instance-level feature adaptation will fall into the dilemma of transferability and discriminability, and do not work well on this difficult dataset. In contrast, our method effectively solves this problem by decoupling the adversarial adaptation from the training of the detector and improves mAP by **7.0** compared with the state-of-the-art.

Table 2: Results from VOC to Comic (ResNet-101). Oracle results are obtained by training on labeled data in the target domain.

| Method | bike | bird | car | cat | dog | prsn | mAP |
|---|---|---|---|---|---|---|---|
| Source Only | 32.5 | 12.0 | 21.1 | 10.4 | 12.4 | 29.9 | 19.7 |
| DA-Faster [8] | 31.1 | 10.3 | 15.5 | 12.4 | 19.3 | 39.0 | 21.2 |
| SWDA [43] | 36.4 | 21.8 | 29.8 | 15.1 | 23.5 | 49.6 | 29.4 |
| MCAR [58] | 47.9 | 20.5 | 37.4 | 20.6 | 24.5 | **53.6** | 33.5 |
| Instance Adapt | 39.5 | 17.7 | 26.5 | 27.3 | 22.4 | 48.4 | 30.3 |
| Global Adapt | 31.9 | 15.7 | 30.3 | 21.3 | 17.1 | 37.9 | 25.7 |
| D-adapt | **52.4** | **25.4** | **42.3** | **43.7** | **25.7** | 53.5 | **40.5** |
| Oracle | 42.2 | 35.3 | 31.9 | 46.2 | 40.9 | 70.9 | 44.6 |

**Adaptation from synthetic to real images.** We use Sim10k as the source domain and Cityscapes as the target domain. Following [43], we evaluate on the validation split of the Cityscapes and report the mAP on *car*. Table 3 shows that our method surpasses all other methods.

Table 3: Sim10k to Cityscapes.

| Method | Backbone | AP on Car |
|---|---|---|
| Source Only | | 34.6 |
| DA-Faster [8] | | 38.9 |
| BDC-Faster [43] | | 31.8 |
| SWDA [43] | | 40.1 |
| MAF [18] | | 41.1 |
| Selective DA [61] | VGG-16 | 43.0 |
| CDN [48] | | 49.3 |
| HTCN* [5] | | 42.5 |
| CFFA [59] | | 43.8 |
| ATF [19] | | 42.8 |
| CADA [20] | | 49.0 |
| MeGA-CDA [52] | | 44.8 |
| UMT* [10] | | 43.1 |
| D-adapt | | **50.3** |
| Oracle | | 69.7 |
| Source-only | | 41.8 |
| CADA [20] | ResNet101 | 51.2 |
| D-adapt | | **51.9** |
| Oracle | | 70.4 |

Table 4: Results from Cityscapes to Foggy Cityscapes.

| Method | Backbone | prsn | rider | car | truck | bus | train | mcycle | bcycle | MAP |
|---|---|---|---|---|---|---|---|---|---|---|
| Source only | | 25.1 | 32.7 | 31.0 | 12.5 | 23.9 | 9.1 | 23.7 | 29.1 | 23.4 |
| DA-Faster [8] | | 25.0 | 31.0 | 40.5 | 22.1 | 35.3 | 20.2 | 20.0 | 27.1 | 27.7 |
| BDC-Faster [43] | | 26.4 | 37.2 | 42.4 | 21.2 | 29.2 | 12.3 | 22.6 | 28.9 | 27.5 |
| SW-DA [43] | | 36.2 | 35.3 | 43.5 | 30.0 | 29.9 | 42.3 | 32.6 | 24.5 | 34.3 |
| Selective DA [61] | VGG-16 | 33.5 | 38.0 | 48.5 | 26.5 | 39.0 | 23.3 | 28.0 | 33.6 | 33.8 |
| DD-MRL* [28] | | 30.8 | 40.5 | 44.3 | 27.2 | 38.4 | 34.5 | 28.4 | 32.2 | 34.5 |
| CADA [20] | | 41.9 | 38.7 | 56.7 | 22.6 | 41.5 | 26.8 | 24.6 | 35.5 | 36.0 |
| CRDA [53] | | 32.9 | 43.8 | 49.2 | 27.2 | 45.1 | 36.4 | 30.3 | 34.6 | 37.4 |
| CFFA [59] | | 34.0 | 46.9 | 52.1 | 30.8 | 43.2 | 29.9 | 34.7 | 37.4 | 38.6 |
| ATF [19] | | 34.6 | 47.0 | 50.0 | 23.7 | 43.3 | 38.7 | 33.4 | 38.8 | 38.7 |
| MCAR [58] | | 32.0 | 42.1 | 43.9 | 31.3 | 44.1 | **43.4** | **37.4** | 36.6 | 38.8 |
| HTCN* [20] | | 33.2 | 47.5 | 47.9 | **31.6** | **47.4** | 40.9 | 32.3 | 37.1 | 39.8 |
| D-adapt | | 43.1 | 51.8 | 58.1 | 26.3 | 36.8 | 14.6 | 32.2 | 42.0 | 38.1 |
| D-adapt* | | **44.9** | **54.2** | **61.7** | 25.6 | 36.3 | 24.7 | 37.3 | **46.1** | **41.3** |
| Oracle | | 47.4 | 40.8 | 66.8 | 27.2 | 48.2 | 32.4 | 31.2 | 38.3 | 41.5 |
| Source-only | | 33.8 | 34.8 | 39.6 | 18.6 | 27.9 | 6.3 | 18.2 | 25.5 | 25.6 |
| CADA [20] | ResNet101 | 41.5 | 43.6 | 57.1 | 29.4 | 44.9 | 39.7 | 29.0 | 36.1 | 40.2 |
| D-adapt | | 42.8 | **48.4** | 56.8 | 31.5 | 42.8 | 37.4 | **35.2** | 42.4 | 42.2 |
| D-adapt* | | 40.8 | 47.1 | **57.5** | **33.5** | **46.9** | **41.4** | 33.6 | **43.0** | **43.0** |
| Oracle | | 44.7 | 43.9 | 64.7 | 31.5 | 48.8 | 44.0 | 31.0 | 36.7 | 43.2 |

**Adaptation between similar domains.** We perform adaptation from Cityscapes to FoggyCityscape and report the results[3] in Table 4. Note that since the two domains are relatively similar, the performance of adaptation is already close to the *oracle* results.

## 4.4 ABLATION STUDIES

In this part, we will analyze both the performance of the detector and the adaptors. Denote $n_{ij}$ be the number of proposals of class $i$ predicted as class $j$, $t_i$ be the total number of proposals of class $i$, and $N$ be the number of classes (including the background), then we use $\text{mIoU}^{\text{cls}} = \frac{1}{N} \frac{\sum_i n_{ii}}{t_i + \sum_j n_{ji} - n_{ii}}$ to measure the overall performance of the category adaptor. We use the intersection-over-union between the predicted bounding boxes and the ground truth boxes, i.e., $\text{mIoU}^{\text{reg}}$, to measure the performance of the bounding box adaptor. All ablations are performed on VOC $\rightarrow$ Clipart and the iteration $T$ is kept 1 for a fair comparison.

**Ablation on the category adaptation.** Table 6(a) show the effectiveness of several specific designs mentioned in Section 3.2. Among them, the weight mechanism has the greatest impact, indicating the necessity of the low-density assumption in the adversarial adaptation. To verify this, we assume that the ground truth IoU of each proposal is known, and then we select the proposal with IoU greater than a certain threshold

Table 5: Effect of proposals' quality.

| IoU threshold | 0.05 | 0.3 | 0.5 | 0.7 |
|---|---|---|---|---|
| mIoU$^{\text{cls}}$ | 36.1 | 38.2 | 46.7 | 51.4 |

when we train the category adaptor. Table 5 shows that as the IoU threshold of the foreground proposals improves from 0.05 to 0.7, the accuracy of the category adaptor will increase from 36.1 to 51.4, which shows the importance of the low-density separation assumption.

**Ablation on the bounding box adaptation.** Table 6(b) illustrates that minimizing the disparity discrepancy improves the performance of the box adaptor and bounding box adaptation improves the performance of the detector in the target domain.

**Ablation on the training strategy with pseudo-labels.** In Equation 2, losses are only calculated on the regions where the proposals are located, and those anchor areas overlapping with the proposals are ignored. Here, we compare this strategy with the common practice in self-training – filter out bounding boxes with low confidence, then label each proposal that overlaps with these boxes. Although the category labels of these bounding boxes are also generated from the category adaptor, the accuracy of these generated proposals is low (see Table 6(c)). In contrast, our strategy is more conservative and both the mIoU$^{\text{cls}}$ on the proposals and the final mAP of the detector are higher.

## 4.5 ANALYSIS

---

[3]* denotes this method utilizes CycleGAN to perform source-to-target translation.

Table 6: Ablations on PASCAL VOC to Clipart. Note that no bounding box adaptation is adopted in (a) and (c) for a fair comparison. **(a)** Category adaptation. *w/o condition*: use a class-independent discriminator. *w/o bg proposals*: no background proposals added to source domain or target domain or neither. *w/o weight*: remove the weight mechanism in Equation 3. *w/o adaptor*: remove the category adaptation step and directly use the labels generated from detector on the target domain as pseudo labels. **(b)** Spatial Adaptation. *w/o DD*: remove the disparity discrepancy in Equation 6. *w/o adaptor*: remove the bounding box adaptation step and only trains the classification branch of the detector. **(c)** Training strategy. In the standard training, if the confidence threshold increases, the number of false negatives will increase, otherwise the number of false positives will increase.

(a) Category adaptation

| metric | ours | w/o condition | w/o bg proposals | | | w/o weight | w/o adaptor |
|---|---|---|---|---|---|---|---|
| | | | source | ✗ | ✗ | ✔ | | |
| | | | target | ✗ | ✔ | ✗ | | |
| mIoU$^{cls}$ | 38.2 | 36.9 | - | 36.6 | 33.6 | 25.1 | 17.2 | 12.6 |
| mAP | 43.5 | 41.7 | - | 41.7 | 38.8 | 36.5 | 33.3 | 28.0 |

(b) Spatial Adaptation

| metric | Ours | w/o DD | w/o adaptor |
|---|---|---|---|
| mIoU$^{reg}$ | 0.631 | 0.598 | 0.531 |
| mAP | 45.0 | 44.4 | 43.5 |

(c) Training strategy

| metric | standard way | | | | ours |
|---|---|---|---|---|---|
| confidence threshold | 0.1 | 0.3 | 0.5 | 0.7 | 0.1 |
| mIoU$^{cls}$ | 17.2 | 17.6 | 17.1 | 16.3 | **38.2** |
| mAP | 38.9 | 37.3 | 35.9 | 34.4 | **43.5** |

**Error Analysis.** Figure 5 gives the percent of error of each model on VOC→Clipart following [2]. The main errors in the target domain come from: *Miss* (ground truth regarded as backgrounds) and *Cls* (classified incorrectly). *Loc* (classified correctly but localized incorrectly) errors are slightly less, but still cannot be ignored especially after category adaptation, which implies the necessity of box adaptation in object detection. Category adaptation can effectively reduce the proportion of Cls errors while increasing that of Loc errors, thus it is reasonable to cascade the box adaptor after the category adaptor. Bounding box adaptation can reduce the proportion of Loc errors, revealing its effectiveness.

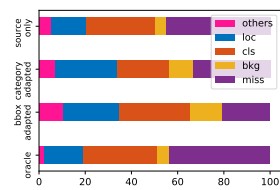

Figure 5: Error analysis.

**Feature visualization.** We visualize by t-SNE [51] in Figures 6(a)-6(b) the representations of task VOC → Comic2k (6 classes) by category adaptor with $\lambda = 0$ and category adaptor with $\lambda = 1$. The source and target are well aligned in the latter, which indicates that it learns domain-invariant features. We also extract box features from the detector and get Figure 6(c)-6(d). We find that *the features of the detector do not have an obvious cluster structure, even on the source domain*. The reason is that the features of the detector contain both category information and location information. Thus adversarial adaptation directly on the detector will hurt its discriminability, while our method achieves better performance through decoupled adaptation.

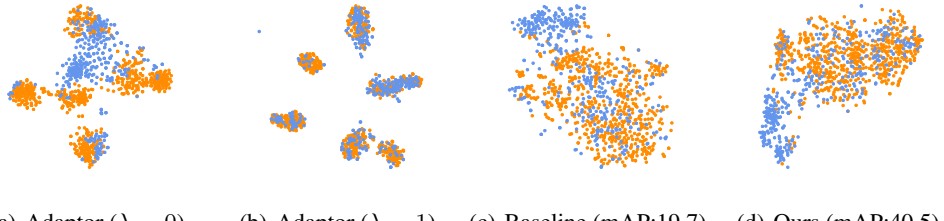

| (a) Adaptor ($\lambda = 0$) | (b) Adaptor ($\lambda = 1$) | (c) Baseline (mAP:19.7) | (d) Ours (mAP:40.5) |

Figure 6: T-SNE visualization of features. (a) and (b) are features from the category adaptor. (c) and (d) are features from the Faster RCNN. (Orange: VOC; Blue: Comic2k).

## 5 DISCUSSION AND CONCLUSION

Our method achieved considerable improvement on several benchmark datasets for domain adaptation. In actual deployment, the detection performance can be further boosted by employing stronger adaptors without introducing any computational overhead since the adaptors can be removed during inference. It is also possible to extend the D-adapt framework to other detection tasks, e.g., instance segmentation and keypoint detection, by cascading more specially designed adaptors. We hope D-adapt will be useful for the wider application of detection tasks.

## ACKNOWLEDGEMENTS

This work was supported by the National Megaproject for New Generation AI (2020AAA0109201), National Natural Science Foundation of China (62022050 and 62021002), Beijing Nova Program (Z201100006820041), and BNRist Innovation Fund (BNR2021RC01002).

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

# A    MORE EXPERIMENT RESULTS

**Results on Other Architecture.**    As shown in Tables 7, our method also applies to the one-stage detector RetinaNet [32] , which improves the mAP by 17.5 on VOC → Clipart.  The proposed D-adapt framework also surpasses both image-level 1(b) and feature-level 1(c) alignment as well as their combination by a considerable margin.

Table 7: Results from PASCAL VOC to Clipart (RetinaNet, ResNet101).

| Method | aero | bcycle | bird | boat | bottle | bus | car | cat | chair | cow | table | dog | hrs | bike | prsn | plnt | sheep | sofa | train | tv | mAP |
|---|---|---|---|---|---|---|---|---|---|---|---|---|---|---|---|---|---|---|---|---|---|
| Source Only | 30.1 | 40.8 | 21.7 | 15.3 | 28.4 | 51.6 | 33.1 | 13.1 | 34.5 | 14.2 | **29.6** | 16.2 | 21.4 | 53.1 | 37.4 | 30.3 | 6.9 | 24.8 | 31.8 | 42.1 | 28.8 |
| Global Adapt | 33.2 | 43.4 | 23.8 | 24.5 | 43.4 | 54.9 | 36.5 | 6.5 | 36.0 | 19.1 | 26.4 | 13.0 | 23.6 | 49.4 | 52.6 | 39.8 | 5.8 | 27.6 | 39.1 | 54.1 | 32.6 |
| Local Adapt | 31.0 | 28.3 | 26.2 | 18.2 | 42.2 | 53.5 | 33.6 | 18.4 | 37.2 | 33.2 | 28.7 | 14.3 | 33.4 | 54.6 | 48.7 | 40.4 | 6.8 | **30.4** | 42.1 | 48.1 | 33.4 |
| Global + Local | 37.5 | 50.4 | 25.3 | 28.8 | 45.0 | 51.7 | 45.9 | 16.9 | 38.2 | 31.9 | 24.2 | 12.6 | 26.4 | 48.7 | 53.4 | 44.5 | 5.5 | 28.2 | 45.7 | 53.5 | 35.7 |
| D-adapt | **47.4** | **65.0** | **33.1** | **37.5** | **56.8** | **61.2** | **55.1** | **27.3** | **45.5** | **51.8** | 29.1 | **29.6** | **38.0** | **74.5** | **66.7** | **46.0** | **24.2** | 29.3 | **54.2** | **53.8** | **46.3** |

**Results on VOC→WaterColor.**    As shown in Table 8, D-adapt also achieves strong performance on WaterColor dataset.

Table 8: Results from VOC to WaterColor (ResNet-101).

| Method | bike | bird | car | cat | dog | prsn | mAP |
|---|---|---|---|---|---|---|---|
| Source Only | 68.8 | 46.8 | 37.2 | 32.7 | 21.3 | 60.7 | 44.6 |
| BDC-Faster [43] | 68.6 | 48.3 | 47.2 | 26.5 | 21.7 | 60.5 | 45.5 |
| DA-Faster [8] | 75.2 | 40.6 | 48.0 | 31.5 | 20.6 | 60.0 | 46.0 |
| WST-BSR [27] | 75.6 | 45.8 | 49.3 | 34.1 | 30.3 | 64.1 | 49.9 |
| MAF [18] | 73.4 | 55.7 | 46.4 | 36.8 | 28.9 | 60.8 | 50.3 |
| SWDA [43] | 82.3 | 55.9 | 46.5 | 32.7 | 35.5 | 66.7 | 53.3 |
| ATF [19] | 78.8 | **59.9** | 47.9 | 41.0 | 34.8 | 66.9 | 54.9 |
| SCL [45] | 82.2 | 55.1 | 51.8 | 39.6 | 38.4 | 64.0 | 55.2 |
| MCAR [58] | 87.9 | 52.1 | 51.8 | 41.6 | 33.8 | 68.8 | 56.0 |
| UMT* [10] | **88.2** | 55.3 | 51.7 | 39.8 | 43.6 | **69.9** | **58.1** |
| D-adapt | 77.4 | 54.0 | **52.8** | **43.9** | **48.1** | 68.9 | 57.5 |
| Oracle | 48.5 | 54.7 | 41.3 | 36.2 | 52.6 | 74.6 | 51.3 |

**Ablations on the decouple strategy.**    Further, we discuss whether the decoupling of different adaptors is useful.

In our original implementation, the input distributions of different adaptors are completely different. In the category adaptation step, we encourage the input proposals to have IoU close to 0 or 1 to better satisfy the low-density separation assumption.  In the bounding box adaptation step, we encourage the input proposals to have IoU between 0.5 and 1 to ease the optimization of the bounding box localization task.

If different adaptors are coupled, they must share the same input distribution.  Table 9 shows that *only sharing the input distributions will greatly damage their respective performance.*  Note that different adaptors still have independent architectures.  And we can conclude that the decoupling of different adaptors is quite crucial.

Table 9: Ablations on the decouple strategy on VOC→Clipart.

| Input Distribution | mIoU$^{cls}$ | mIoU$^{reg}$ |
|---|---|---|
| all proposals w/o weight (both adaptors use the proposals directly output by the detector) | 17.2 | 0.551 |
| all proposals w/ weight (both adaptors use the proposals fed to the original category adaptor) | **33.3** | 0.319 |
| foreground proposals weight (both adaptors use the proposals fed to the original box adaptor) | 24.7 | **0.631** |
| Ours (different adaptors have different input data distributions) | **33.3** | **0.631** |

Table 10: Ablations on the box adaptor when $T$ varies.

| Setting | mAP (T=1) | mAP (T=2) | mAP (T=3) |
|---|---|---|---|
| without box adaptor | 43.5 | 45.8 | 47.0 |
| with box adaptor (ours) | 45.0 | 47.7 | 49.1 |

**Ablation on bounding box adaptor.** Table 10 shows that the gain brought by box adaptation is consistent, for example when $T = 3$, it can still improve the mAP from 47.0 to 49.1.

## B MORE VISUALIZATION RESULTS.

Figure 7-10 gives more qualitative results on Faster RCNN.

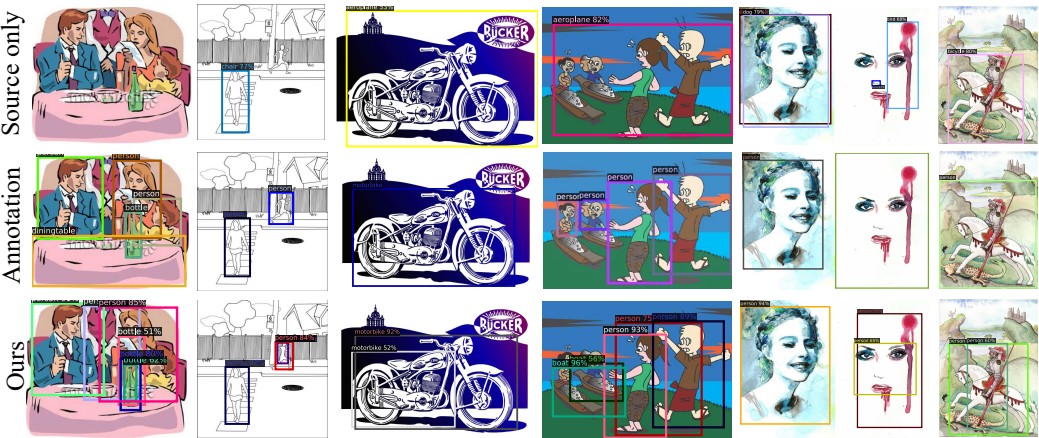

Figure 7: Qualitative results on VOC → Clipart.

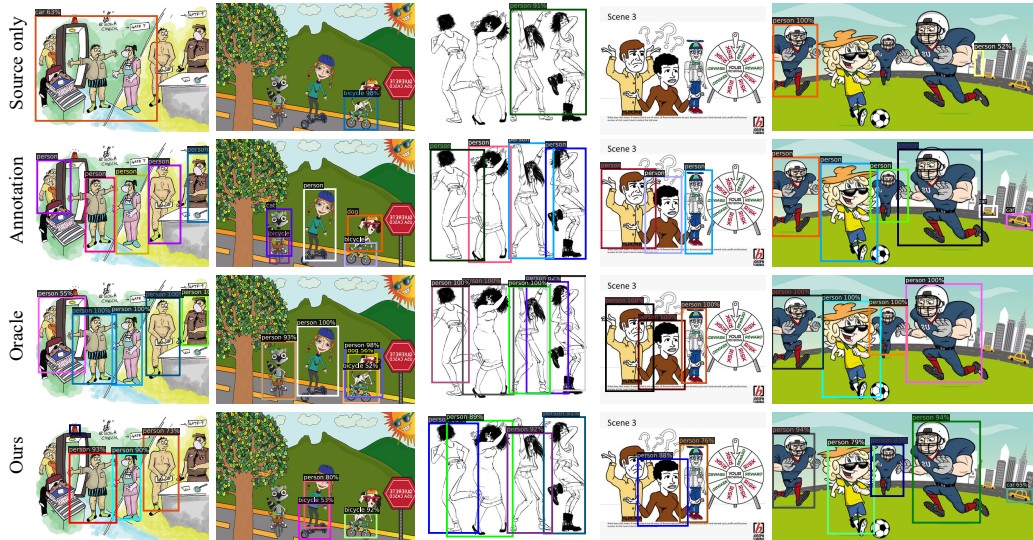

Figure 8: Qualitative results on VOC → Comic.

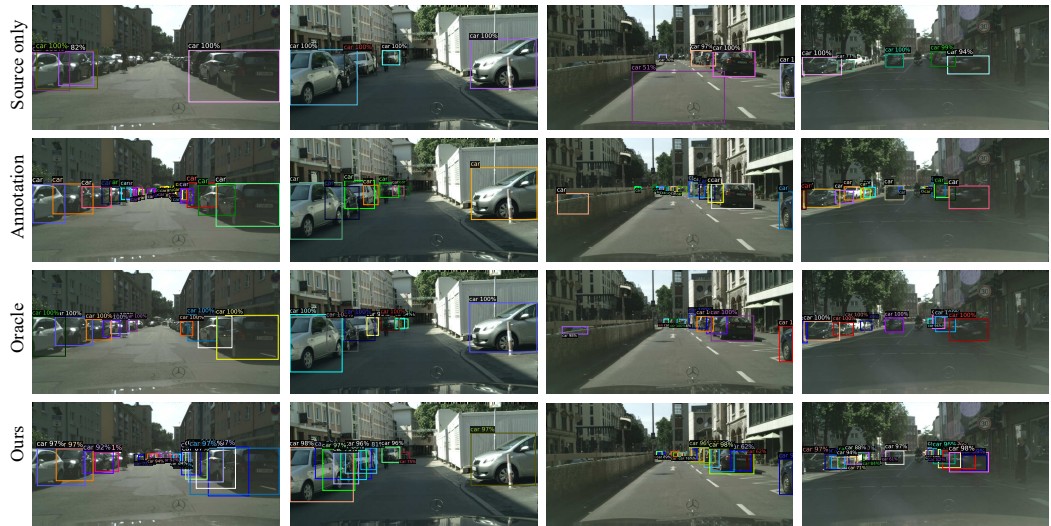

Figure 9: Qualitative results on Sim10k → Cityscapes.

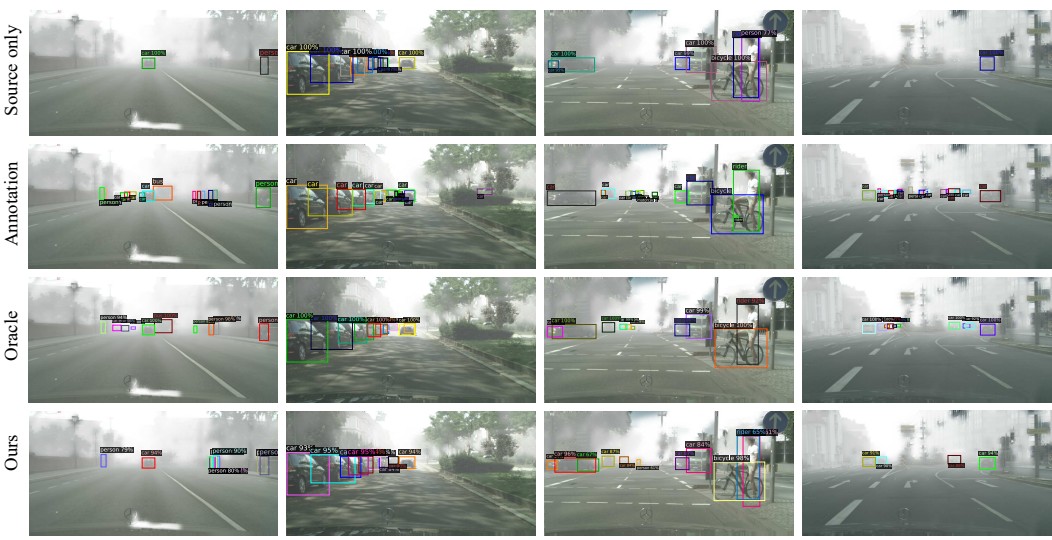

Figure 10: Qualitative results on Cityscapes → Foggy Cityscapes.

