# OpenReview forum: "Decoupled Adaptation for Cross-Domain Object Detection"
_ICLR.cc/2022/Conference — ICLR 2022 Poster_

### Official Review · Reviewer_Kz22 · 2021-11-02

**Correctness:** 3
**Technical Novelty And Significance:** 2
**Empirical Novelty And Significance:** 3
**Recommendation:** 8
**Confidence:** 3

**Main Review:**

Positives:
+ The paper is well written and the problem is well motivated.
+ The experiments on the standard benchmarks are reported and comparisons are provided with recent works.
+ The ablation study is formulated to reflect the claims.

Clarifications needed:
- the bounding box adaptors take as input the foreground proposals, which are the crops of objects, as shown in figure 3. How does the regressor predict offsets if it has been given a close crop of the objects? The claim made is that the confirmation bias of the pseudo labels with the use of the adaptors. If the adaptors are trained to align the features such that the discriminator is confused, which means the source bounding box distribution needs to be matched with the target, hence biasing the regressor again.
-T is set to be 3, is saturation seen after a point. Would different domain pairs have different saturation iteration? How was this value chosen? Shouldn't it be fair to also report mAP scores for T=1, in Table 3 and 4 .
-Table 1 is missing UMT[10] baseline which reports mAP 44.1 for VOC to Clipart which is a stronger baseline for the pseudo-label based techniques.
-Table2 doesn't mention the backbone used in the experiment
- The error analysis part should also include the bounding box adaptor part to highlight how much bias is removed using bounding box adaptor.
- the line on page 9: "The gain brought by box adaptation is consistent, for example when T = 3, it can still improve the mAP from 47.0 to 49.1." should it be 45.0 instead of 47.0?

**Summary Of The Paper:**

This work provides a systematic way to adapt object detectors to an unseen target domain. Specifically, the object classification and bounding box regression heads are adapted one after the other, and ROIs are used to train a separate set of adaptors to provide foreground feature alignment without hurting the discriminablity of the detectors.

**Summary Of The Review:**

The approach highlights the shortcomings of the previous methods and systematically tries to resolve them. There are a few issues with the comparisons and clarification is needed for one of the claims.  Overall the writing and structure of the paper is good.

---

> ### Author Response · Authors · 2021-11-14
> **Response to Reviewer Kz22**
>
> We appreciate the insightful suggestions from Reviewer Kz22. We have clarified the questions in the following feedback.
>
> **Q1:** How to train the bounding box adaptors?
>
> First, the input of the bounding box adaptor (the crops of objects) will be twice larger than the original predicted box, so that the bounding box adapter could access more location information. The exact number of 2 times is not important, since changing to 1.5 times and 2.5 times has a minor effect on the final results.
>
> Second, the predicted boxes on the source domain are still not accurate, especially when the corresponding confidences are low. Therefore, these noisy predictions and their ground truth bounding boxes can provide supervision to the bounding box adaptor.
>
> **Q2:** How to choose T?
>
> Different domains have different saturation points. For example, when the domain difference is large, T will be larger, and when the domain difference is small, T will be smaller. For example, the model trained on Sim10k->Cityscapes is close to saturation when T=1 or 2. Here are the results of  Sim10k->Cityscapes.
>
> | Backbone  | mAP(T=1)  | mAP(T=2)  | mAP(T=3)  |
> |-----------|------|------|------|
> | VGG16     | 48.5 |49.8 |50.3 |
> | ResNet101 | 50.8 |52.5 |53.2 |
>
> In contrast, when the domain discrepancy is large, such as VOC->Clipart, models will saturate when T=3.
>
> | Backbone  | mAP(T=1)  | mAP(T=2)  | mAP(T=3)  |
> |-----------|------|------|------|
> | ResNet101 | 45.0 | 47.7 |49.1 |
>
> Since there are no labeled data on the target domain, we have no way to judge when the modes will saturate. Thus we take T=3 for all datasets.
>
> The reason for using multiple iterations is to increase interaction between the detector and the adaptors, which is an important part of our method. Besides, our training is also faster than many existing domain adaptation methods for object detection.
>
> The following table is a comparison of training time. SWDA [42] trains for 70k iterations, which costs ~18h. In contrast, the training time of our method when T=1 is ~7.5h, and the training time when T=3 is ~16.5h (source domain pre-training requires 3h). Thus, such  a comparison is also fair.
>
> | Method | Training Time  |
> |-----------|------|
> | SWDA [42] | ~18.0h|
> | Ours (T=1) | ~7.5h|
> | Ours (T=2) | ~12h|
> | Ours (T=3) | ~16.5h|
>
>
> **Q3:** UMT [10] is missing in $\underline{\text{Table 1}}$.
>
> UMT [10] takes a different split of Clipart from most of the other work, and thus the comparison with it is not fair. To make a direct comparison with more papers, we adopted the settings in [42, 19, 56].
>
> **Q4:** $\underline{\text{Table 2}}$ doesn't mention the backbone used in the experiment.
>
> The used backbone is ResNet-101. We have corrected this writing mistake in the revised draft.
>
> **Q5:** The error analysis should  include the bounding box adaptor part.
>
> We have added this comparison in $\underline{\text{Figure 5}}$ of the revised draft.
>
> **Q6:** The line on page 9 should be 45.0 instead of 47.0?
>
> We have rechecked the results and found that they are correct in the original version.
>
> For brevity, $\underline{\text{Table 6}}$ only shows the ablation of different modules when T=1. Below is the ablation of the box adaptor when T varies.
>
> | Setting        | mAP (T=1) | mAP (T=2) | mAP (T=3) |
> |-----------------|------|------|------|
> | without box adaptor | 43.5 | 45.8 |47.0 |
> | with box adaptor (ours)  | 45.0 | 47.7 |49.1 |

---

> > ### Comment · Reviewer_Kz22 · 2021-11-18
> > **Response to Authors**
> >
> > Thank you for the clarifications and for providing additional details. Here are my further comments:
> >
> > - My concern regarding the claim of avoiding confirmation bias due to pseudo-labels, still exists. Since the loss formulation of the adaptors utilizes the adversarial training, it would be promoting alignment of the source and target distribution. Hence, bias seems to be inevitable. Could you please clarify how this bias is removed in your formulation?
> > - The additional error analysis shows that the proportion of the mislocalization increases after the bbox adaptation. This seems to be counter-intuitive. Could you please clarify this?

---

> > > ### Author Response · Authors · 2021-11-19
> > > **Response to Reviewer Kz22**
> > >
> > > Thank you for the responsive reply. We answer your additional questions as follows.
> > >
> > > - Confirmation bias is an obstacle mainly existing in pseudo-labeling: the performance of a student is restricted by the teacher when learning from inaccurate pseudo-labels [63]. However, in domain adversarial training, no explicit pseudo-labeling is used to train the model, which means that as the training proceeds, the model can better correct its predictions on the target domain.
> > > Still, confirmation bias cannot be completely avoided, thus our statement is **alleviating** rather than **avoiding** the confirmation bias.
> > >
> > > - We have rechecked the results and found that the additional error analysis is correct. The proportion of the mislocalization **decreases** from 26.8% to 24.5% after the bbox adaptation.
> > >
> > > W'd be very happy to answer any further questions.
> > >
> > > [63] Arazo, E., Ortego, D., Albert, P., O’Connor, N. E., and McGuinness, K. Pseudo-labeling and confirmation bias in deep semi-supervised learning, 2020.

---

> > > > ### Comment · Reviewer_Kz22 · 2021-11-23
> > > > **Response to Authors**
> > > >
> > > > Thank you for the clarifications.
> > > > I would recommend for the sake of completeness, please the discussion on T as done above in the appendix section in the final version.

---

> > > > > ### Author Response · Authors · 2021-11-24
> > > > > **Response to Reviewer Kz22**
> > > > >
> > > > > We'd like to thank Reviewer Kz22 again for providing an impressive valuable pre-rebuttal review and re-clarifying the review with more detailed explanations.
> > > > >
> > > > > We'd also thank you for carefully judging our feedback and raising the score! Your constructive suggestions are very helpful for us to improve the completeness of experiments. We will add the discussions above to our appendix section in our final version.

---

### Official Review · Reviewer_LmKN · 2021-11-03

**Correctness:** 4
**Technical Novelty And Significance:** 3
**Empirical Novelty And Significance:** 4
**Recommendation:** 8
**Confidence:** 3

**Main Review:**

Strengths:
1. The paper is very well-written.
2. The proposed idea is simple and intuitive, but turns out to be quite effective.
3. The decoupled adaption is a great option for avoiding the hurt to the detector's discriminability.
4. The decoupled adaption design enables the ease of incorporating the box regression adaption, which is novel.
5. The experiments are extensive and persuasive.

Weaknesses:
Will the authors release their code? They do not mention it at all.

**Summary Of The Paper:**

The authors propose a tailored method, named Decoupled Adaptation (D-adapt), for cross-domain object detection. The conventional domain adaptation techniques hurt the discriminability of the detector and ignores the adaptation on bounding box regression. In order to avoid hurting the discriminability of the detector, the authors propose to decouple the adaptation modules from the detector, which achieves the parameter independence. In addition, the authors propose the module for adapting the bounding box regression. Due to the decoupling property, the proposed D-adapt framework is compatible to many off-the-shelf detectors. The experimental results are very promising.

**Summary Of The Review:**

Considering the simplicity, novelty and effectiveness, this paper is of high quality and above the acceptance bar.

---

> ### Author Response · Authors · 2021-11-14
> **Response to Reviewer LmKN**
>
> Many thanks for the recognition of our work from Reviewer LmKN.
>
> **Q1:** Will the authors release their code?
>
> Of course yes. As mentioned in $\underline{\text{Section 4.2}}$, **we guarantee to make the code public when published**. Besides, we will release the code to reproduce the baseline results, such as Source Only, instance-level alignment methods, global-level alignment methods, and local-level alignment methods in a **clean implementation**. We hope to benefit our community with high-quality open source project.

---

### Official Review · Reviewer_HwgG · 2021-11-03

**Correctness:** 3
**Technical Novelty And Significance:** 3
**Empirical Novelty And Significance:** 4
**Recommendation:** 8
**Confidence:** 4

**Main Review:**

The quality of the introduction is inferior to the rest of the paper, thus it can be improved.
- As the paper presents the problem, it poses a "Data challenge: what to adapt in the adversarial training is unknown." It is not clear at this point of the paper what is the relation between domain adaptation and adversarial training. For clarity, the connection that currently is made  on "Related Work" should be presented as part of the motivation presented in the introduction to situate the reader on the challenges presented.
- The text poses "Architecture challenge" but it is also not clear on the following text, that should describe the challenge, how the argument is any related to architecture. The arguments presented in the text that follows (again without much context at this point) is that they are related to training policy and regularization, but not architecture specifically. Similar to the first challenge presented, the text is lacking a better context and justification.
- The "loss challenge" will only be clear from section 3.1 on, as the introduction text defining it is not very informative.

For the sake of completeness and clarity, the main concepts related to domain adaptation and adversarial training, such as adaptors for domain adaptation, adversarial alignment and adversarial regressor, must be defined and referenced before its use and/or reformulated in the context of the paper.

On results comparison, it would be important to state any architecture difference (or state otherwise) between the proposal and compared approaches. Do they use same detector architecture?

It is also important to clarify what is the oracle used (and better justify its errors from Figure 5).
In that direction, the text should also clarify the results presented that surpassed the oracle results. This happens on Table 2 for "bike" and "car", and on Table 4, it is not clear how performance on "rider" and "bicycle" surpassed the oracle results for VGG backbone and "truck", "motorcycle" and "bicycle" for Resnet101.

Overall, the experimental results are compelling, but it is necessary to better link the concepts used with pre-existent approaches on adversarial domain adaptation.

**Summary Of The Paper:**

The paper focus on object detection domain adaptation to transfer a detector from a source domain, in the context where sufficient training data is available on a source domain but only unlabeled data is available in the target domain. It presents good results adapting from/to Pascal, Clipart, Comic, Sim10k, Cityscapes and FoggyCityscapes.


**Summary Of The Review:**

The paper presents a clever combination of domain adversarial training for adapting the components of object detection. The experimental results obtain SOTA for some combinations explored. The weakest part is of the presentation is the introduction, and should be reviewed. Also,  although the set of references included have a good cover of the area, the concepts presented in the text are not connected to pre-existent concepts as it would be appropriate to be.

---

> ### Author Response · Authors · 2021-11-14
> **Response to Reviewer HwgG**
>
> We appreciate the detailed advice on the writing from Reviewer HwgG.
>
> **Q1:** The writing of the Introduction can be improved.
>
> Thanks for your useful advice! We have modified the Introduction part:
> - We remove *adversarial training* or other demanding concepts from the Introduction.
> - We have enriched the description of the three challenges to make them clear within the context.
>
> Apologies for the confusion caused by the word *architecture challenge*. Its meaning is the challenge when we integrate domain adaptation methods (DANN, CDAN etc) into the complex object detection architectures (Faster R-CNN, SSD etc).
>
> Previous methods usually introduce some new modules inside a specific detector, leading to an overall change of the architecture. In contrast, our modifications to the architecture are limited to the behaviors of the object detectors --- the input, the output, and the connection between different modules --- without changing the architecture itself. This allows our method to scale easily to different detection architectures.
>
> **Q2:** Any architecture difference?
>
> No. As mentioned in $\underline{\text{Section 4.2}}$, we use the architecture following the previous paper. Specifically, in $\underline{\text{Tables 1 and 2}}$, we report results on Faster RCNN+ResNet101 following [42]. In $\underline{\text{Tables 3 and 4}}$, we report results on both Faster RCNN+VGG16 and Faster RCNN+ResNet101 following [19].
>
> As we mentioned in our response to **Reviewer PJVS/Q6**, some works that adopt a different experiment setting or different architectures are not included for a fair comparison.
>
> **Q3:** What are the oracle results?
>
> The oracle results are obtained by training on labeled data in the target domain following [42, 19，56]. We have added the notes for oracle and justified its errors in $\underline{\text{Figure 8, 9, 10}}$.
>
> Note that, the category distributions on the source and target domains are also different, thus it's very likely that the results of an adapted detector surpass the oracle results on some of the object categories.

---

### Official Review · Reviewer_PJVS · 2021-11-06

**Correctness:** 3
**Technical Novelty And Significance:** 2
**Empirical Novelty And Significance:** 3
**Recommendation:** 6
**Confidence:** 4

**Main Review:**

> Pros:
- The approach is relatively simple and easy to understand
- Performance of the proposed approach is strong, improving on the state-of-the-art

> Cons:
- Despite relative simplicity of the approach, writing obfuscates some of the details, making it more difficult to understand than necessary. Some improvements in exposition, particularly in Section 3, would improve overall readability.
- Technical novelty of the proposed approach is relatively minor. Both adversarial feature adaptation and adaptive refinement of Faster RCNN heads are explored techniques. While the specific strategy proposed in the paper for doing so is novel, the core leveraged technical fundamentals are not. The novelty appears mostly in the use of proposal weighting within adaptation and sequential, rather then parallel, adaptation procedure for the two heads.
- Paper overclaims its contributions. Specifically, it is claimed that "previous methods mainly explored the category adaptation and ignored the regression adaptation" (also Claim 2 of the contributions). This isn't really true. While I am not an expert on cross-domain object detection, a quick search reveals that regression adaptation has been utilized in other past works, e.g., for example in [23] see "Bounding Box Refinement", the pseudo-labeling approaches (e.g., "Multi-Source Domain Adaptation for Object Detection", Yao et al, ICCV) also refine classification and detection. Hence, this claim is at least miss-characterized, needs to be refined and overall contributions clarified.
- For completeness, related work should also discuss approaches for object detection adaptation in semi-supervised learning (not specifically designed for cross-domain object detection, but leveraging techniques relevant to the approach). These include: (1) "Note-rcnn: Noise tolerant ensemble rcnn for semi- supervised object detection", Gao et al, ICCV, 2019 and (2) UniT: Unified Knowledge Transfer for Any-shot Object Detection and Segmentation, Khandelwal et al, CVPR 2021, among others.
- Ablations show significance of the proposal weighting in the adaptation (good!), as well as significance of adaptation of regression for bounding boxes (also good!), however, fail to illustrate that decoupling and sequential nature of adaptors (Figure 1d) is important. Given that this is one of the core contributions, addition of such ablation would be very useful.
- Minor: It is unclear why certain approaches were cited in related work, but do not appear to be compared against in the tables (e.g., [57])

**Summary Of The Paper:**

Paper addresses the problem of cross-domain object detection. The formed source domain detectors (in the form of Faster R-CNN), learned in a supervised manner, are adopted to perform well on the target domain where no annotations are available. The key to the approach is separate adaptation of the classification head and the regression head within Faster RCNN. This leads to the proposed D-adapt, namely Decoupled Adaptation, that decouples the adversarial adaptation and the training of the detector. In addition, classification and detection heads are adopted in tandem. Experiments show that the proposed D-adapt strategy achieves state-of-the-art results on four cross-domain object detection tasks.

**Summary Of The Review:**

Overall, the approach is sounds and experimentally performs well. However, the novelty and significance are somewhat limited. Further, exposition can be improved and claims (specifically claim 2) are (at least) imprecise and miss-characterize prior work. Finally, some ablations that would have been useful are missing and should be included. Given these concerns, overall, I view the paper as being slightly below bar for acceptance. I would be happy to revisit my assessment, and potentially increase the score, if authors can convincingly address the raised concerns in the rebuttal.

Authors have addressed a number of my comments in the rebuttal, and promised to addressed others (claims) in the camera ready revision. As such, I am raising my score and believe the paper should be accepted.

---

> ### Author Response · Authors · 2021-11-14
> **Response to Reviewer PJVS**
>
> We appreciate the thorough review from Reviewer PJVS. We have clarified the questions in the following response.
>
> **Q1:** Some improvements in exposition.
>
> To make our paper more readable, we invited some researchers and students with different expertise to read our paper and revised it based on their feedbacks until they can understand our work. The updated part is highlighted in blue font in the revised draft.
>
> In the Method part, we add a brief introduction to some closely related works on domain adaptation and object detection. This will make all concepts clear from the context. Due to the limited space, we were not able to introduce these methods in detail. Therefore, some background knowledge in object detection and domain adaptation is still necessary for a complete understanding of this paper.
>
> **Q2:** The novelty of the proposed method.
>
> Our novelty includes three aspects.
>
> - We propose a novel **decouple strategy**, which is based on our *rethinking* of the necessity for the object detector to incorporate adaptation modules.   While previous work tried various methods to fuse the detector with the  adaptation modules, such as applying adversarial training on different levels of features of the detector or introducing complicated attention or memory mechanisms into the adaptation module, in experiments we find that incorporating adaptation modules *inside the detectors* might lead us to a wrong direction towards cross-domain object detection. As shown in $\underline{\text{Figure 6}}$, we find that the features of the detector do not have an obvious cluster structure, even on the source domain, since the features of the detector contain both category information and location information. Thus adversarial adaptation directly on the detector will hurt the discriminability of representations, while our method achieves much better performance through the proposed decoupled adaptation.
>
> - Besides the novel decouple strategy, we find that proposals that are neither foreground nor background are harmful to category adaptation. Thus, we propose a novel **unsupervised discretization method** in $\underline{\text{Section 3.2}}$.
>
> - Since domain adaptation for arbitrary regression problems remains an under-explored problem, we propose a brand new **IoU disparity discrepancy method** to deal with object localization tasks under a cross-domain setting $\underline{\text{Section 3.3}}$.
>
> Overall, cross-domain object detection is a **hard problem** because directly combining domain adaptation with object detectors cannot yield strong results. We believe that delving into this hard problem by uncovering **nontrivial insights that lead to a solid solution of strong performance** is also very valuable to our community.

---

> > ### Author Response · Authors · 2021-11-14
> > **Response to Reviewer PJVS**
> >
> > **Q3:** Overclaim contributions on regression adaptation.
> >
> > Apologies for the misunderstanding caused by this sentence and we have modified the argument in question. Foremost, we clarify that **"regression adaptation"** in our paper refers to deep regression across domains of different distributions by the adaptation of different domains into a similar distribution.
> >
> > Although both regression adaptation and semi-supervised regression aim at generalizing from the labeled samples to the unlabeled samples, we need to clarify that they are different in their basic assumptions. **Semi-supervised regression** assumes the labeled and unlabeled samples come from the same or similar distributions and thus they do not have to consider closing the distribution shift in their methods. When the distributions of unlabeled data and labeled data are identical or relatively close, this might work. Yet when the domain discrepancy enlarges, the performance of semi-supervised regression cannot be guaranteed either theoretically or empirically.
> >
> > - For instance, [23] is based on semi-supervised regression methods since it generates pseudo labels from weak augmentation of images and uses these labels to refine the predictions from strong augmentation. [23] only gives results on similar domains, such as Cityscapes->FoggyCityscapes. However, the performance is not guaranteed when the domain discrepancy is large. Below is the comparison between this semi-supervised regression method with our proposed regression adaptation method on the object localization task of VOC->Clipart, which is of large domain discrepancy.
> >
> >     | Method  | Performance of box adaptor $\text{mIoU}^\text{reg}$ |
> >     |-----------|------|
> >     | Source Only | 0.598 |
> >     | Semi-supervised regression in [23] | 0.601 |
> >     | IoU Disparity Discrepancy (Ours) | 0.631  |
> >
> > - "Multi-Source Domain Adaptation for Object Detection" adds a consistency regularization between the target detector and the detector obtained from each source, which is limited to the scenarios where multiple sources exist. Also, it does not solve the core problem of domain adaptation to narrow the distribution discrepancy.
> >
> > Domain adaptation of arbitrary regression tasks between two distributions with large discrepancy is still a difficult problem [7, 21]. To the best of our knowledge, there are no regression adaptation methods for object localization tasks. Nevertheless, we follow the reviewer's suggestion and expand our discussion to semi-supervised regression methods.
> >
> > **Q4:** Related work on semi-supervised object detection.
> >
> > Besides the Unbiased Teacher cited in our original version, we will add citations in the following paragraph. Temporarily, we do not add this paragraph into the revised draft to keep the citation numbers unchanged, which are used in both the reviews and our replies.
> >
> > "When some image-level labels exist, the performance can be further improved by encoding correlations between coarse-grained and fine-grained classes [60], employing noise-tolerant training strategies [61], or learning a mapping from weakly-supervised to fully-supervised detectors [62]."
> >
> > [60] 2019 CVPR Detecting 11K Classes: Large Scale Object Detection without Fine-Grained Bounding Boxes
> >
> > [61] 2019 ICCV NOTE-RCNN: NOise Tolerant Ensemble RCNN for Semi-Supervised Object Detection
> >
> > [62] 2021 CVPR UniT: Unified Knowledge Transfer for Any-shot Object Detection and Segmentation

---

> > > ### Author Response · Authors · 2021-11-14
> > > **Response to Reviewer PJVS**
> > >
> > > **Q5:** Ablations on the decouple strategy.
> > >
> > > 1. $\underline{\text{Table 2 and Table 7}}$ present the difference between the decouple strategy and non-decouple strategy (such as Instance Adapt, Global Adapt, Local Adapt) under different detection architectures. It reveals that the decoupling between the detector and the adaptors is very important.
> > >
> > > 2. Further, we discuss whether the decoupling of different adaptors is useful. Note that in our original implementation, the input distributions of different adaptors are completely different.
> > >    - In the category adaptation step, we encourage the input proposals to have IoU close to 0 or 1 to better satisfy the low-density separation assumption.
> > >    - In the bounding box adaptation step, we encourage the input proposals to have IoU between 0.5 and 1 to ease the optimization of the bounding box localization task.
> > >    - Note that, if different adaptors are coupled, they must share the same input distribution.
> > >
> > >    Here we provide the following results on VOC->Clipart (the setup is the same as that in $\underline{\text{Section 4.4}}$). Note that different adaptors still have independent architectures. **Yet only sharing the input distributions will greatly damage their respective performance**. Thus, we can conclude that the decoupling of different adaptors is quite crucial.
> > >
> > >    | input distribution       | $\text{mIoU}^\text{cls}$ | $\text{mIoU}^\text{reg}$ |
> > >    |--------------------------|---------|---------|
> > >    | all proposals w/o weight (both adaptors use the proposals directly output by the detector) | 17.2    |   0.551      |
> > >    | all proposals w/ weight (both adaptors use the proposals fed to the original category adaptor) | **33.3**    |   0.319      |
> > >    | foreground proposals weight (both adaptors use the proposals fed to the original box adaptor)   | 24.7       | **0.631**   |
> > >    | Ours (different adaptors have different input data distributions)    | **33.3**      | **0.631**   |
> > >
> > > **Q6:** Why are certain approaches only in related work, not in the tables?
> > >
> > > As per your suggestion, we add some missing results to the revised draft. Note that our method still surpasses these baselines sharply.
> > >
> > > Note that the remaining work adopts a different experiment setting from other works, e.g., UMT [10] takes a different split of Clipart, MTOR [3] and Robustness [23] adopt different backbones (ResNet-50 and Inception-v2 respectively).

---

> > > > ### Comment · Reviewer_PJVS · 2021-11-24
> > > > **Response to rebuttal**
> > > >
> > > > I have read the responses from the authors and have a few additional comments.
> > > >
> > > > 1. I found response to Q5.2 compelling. It should be added to the main paper or supplementals.
> > > >
> > > > 2. You call the proposed approach in Section 3.2, "unsupervised discretization method". I do not see why. There is absolutely nothing "discrete" about the proposed approach. "Discretizing the input space", as is stated in the paper, would imply breaking the inputs into sets. This is not what happens. As far as I can see the weight is a continuous function of proposal confidence (Eq. 3), so there is no discretization that takes place. A strategy of weighting loss/error by the confidence is a fairly standard technique in ML. The use of it in this context is clever, but I would not consider this a significant technical innovation.
> > > >
> > > > 3. Generally, I do agree with the authors that the problem being addressed is "a hard problem". I also agree that there is clear novelty in the proposed approach. I realize that level of novelty is difficult to objectively measure and any arguments about it, on both sides, are fundamentally subjective. I still believe the technical novelty is somewhat limited, but concede that the approach is valuable and I will not oppose acceptance on these grounds.
> > > >
> > > > 4. Perhaps most importantly, I find the following statement "Foremost, we clarify that "regression adaptation" in our paper refers to deep regression across domains of different distributions by the adaptation of different domains into a similar distribution." very confusing. Further, the paper still claims: "We propose an effective method to adapt the bounding box localization task, which is ignored by existing methods but is crucial for achieving superior final performance.". While I agree with the first part of the statement: "We propose an effective method to adapt the bounding box localization task". I believe the second part is incorrect, i.e., "is ignored by existing methods". As I point out in my review other methods do perform adoptive regression. They are not doing it in the same way, true, but they are also not completely "ignoring it". If the authors are willing to fix this claim, I would be happy to upgrade my score and more actively argue for acceptance.

---

> > > > > ### Author Response · Authors · 2021-11-25
> > > > > **Response to Reviewer PJVS**
> > > > >
> > > > > Thank you for the responsive reply with additional comments. We answer your additional questions as follows.
> > > > >
> > > > > 1. We're pleased with your recognition. We will add the experiments in Q5.2 to our future version.
> > > > >
> > > > > 2. Apologies for the confusion caused by the word *"discretization"*. We further explain it as follows:
> > > > >
> > > > > - In supervised object detection, it's common to convert the continuous IoUs into discrete positive (IoU>0.7) and negative (IoU<0.3) labels, and then use them for training. This is termed as "discretization" in our paper.
> > > > >
> > > > > - In cross-domain object detection with unlabeled data in the target domain, we also want to use this "discretization" strategy. Yet we don't know the ground truth IoU, thus we use the confidence-based weight to improve the probability that the proposal participating in adaptation is a positive or negative example.
> > > > >
> > > > > - We will change the use of "discretization" into "IoU thresholding" in our future version to avoid similar confusion.
> > > > >
> > > > > 3. Again, thank you for your recognition on the contribution of our paper.
> > > > >
> > > > > 4. We find this suggestion constructive and will fix our claim in our revision. Concretely, we will remove the statement of "is ignored by existing methods" throughout the paper. We will position our approach as a new technique to adaptive regression in cross-domain object detection.

---

### Author Response · Authors · 2021-11-14
**Summary of Revisions**

We appreciate all four reviewers for their insightful and constructive comments.
We have uploaded a revised draft to address all reviewers' comments. Below is a summary of the main changes:

- We have modified the Introduction part to make the meanings of some concepts clearer while avoiding the usage of too many domain adaptation concepts.

- In the Method part, we add a brief introduction to previous works of prerequisite concepts that are closely related to the concepts we will use.

- Based on the reviewers’ comments, we add more experimental details and results.

We hope our responses and revisions will address all reviewers’ concerns!

---

### Decision · Program_Chairs · 2022-01-20

**Decision:**

Accept (Poster)

**Comment:**

This paper decouples the adversarial training of a domain adaptation model with the detector learning process, and is able to disentangle the features of foreground and background when performing adaptation.  State of the art results on four different domains/tasks are presented with significant improvement. Reviewers are unanimous that the submission is acceptable.  Reviewer PJVS is the most authoritative and experienced reviewer, and notes the paper clarity is impaired, and that the paper is immodest in various places and overclaims what is otherwise supported by the results therein.  The AC concurs with this view.  However the paper is acceptable in present form for ICLR, and the AC advises the authors to revise as discussed in the rebuttal and response to the rebuttal.